

# Exploratory analyses of the associations between Ki-67 expression, lymph node metastasis, and prognosis in patients with esophageal squamous cell cancer

Jianqing Zheng[1,*], Bifen Huang[2,*], Ying Chen[1], Bingwei Zeng[3], Lihua Xiao[1] and Min Wu[1]

[1] Department of Radiation Oncology, The Second Affiliated Hospital of Fujian Medical University, Quanzhou, Fujian, China
[2] Department of Obstetrics and Gynecology, Quanzhou Medical College People's Hospital Affiliated, Quanzhou, Fujian, China
[3] Department of Pathology, The Second Affiliated Hospital of Fujian Medical University, Quanzhou, Fujian, China
[*] These authors contributed equally to this work.

Corresponding author
Min Wu, wumin8303@163.com

## ABSTRACT

**Background**. The relationships between Ki-67/MKI67 expression, lymph node metastasis (LNM), vascular invasion (VI), and perineural invasion (PI) in esophageal squamous cell cancer (ESCC) remain unclear. This retrospective cohort study was performed to evaluate the prognostic value of Ki-67 expression and its association with LNM in patients with resected ESCC.

**Methods**. The analysis included 168 patients with ESCC with available Ki-67 protein expression data. The patients were divided into Ki-67 high-expression group (Ki-67 High, 93 cases) and Ki-67 low-expression (Ki-67 Low, 75 cases) groups. Associations between Ki-67 expression and ESCC pathological features was assessed using chi-square test. Overall survival (OS) was compared between the two groups using Kaplan–Meier survival analysis and Cox proportional hazards model.

**Results**. Median follow-up duration was 33.5 months (range 3.0–60.0 months). High Ki-67 expression was significantly associated with poor OS in patients with ESCC compared to that of the low-expression in both univariate (hazard ratios (HR) = 3.42, 95% CI [2.22–5.27], $P < 0.001$) and multivariate analyses (HR = 1.98, 95% CI [1.33–2.94], $P < 0.001$). Furthermore, high Ki-67 expression was significantly associated with an increased risk of LNM ($\chi^2 = 11.219$, $P = 0.011$), VI ($\chi^2 = 6.359$, $P = 0.012$), and PI ($\chi^2 = 8.877$, $P = 0.003$).

**Conclusions**. High Ki-67 protein expression is associated with poor prognosis in ESCC. Increased Ki-67 expression significantly increases the risk of LNM, VI, and PI in ESCC, and thus may serve as an indication for adjuvant therapy in ESCC management.

## INTRODUCTION

Esophageal cancer (EC) is a prevalent and often fatal malignancy worldwide. The incidence and mortality rates of EC in China are considerably higher than the global average for both developed and developing countries. EC ranks fourth in cancer-related mortality rate in both urban and rural areas (*Zhu et al., 2023*). Notably, China accounts for over 50% of global cases, with particularly severe in vulnerable populations (*Li et al., 2021a*). The primary pathological subtype of EC in China is esophageal squamous cell carcinoma (ESCC), accounting for 80% of cases. This prevalence differs from that in developed countries such as the United States (*An et al., 2023*; *Liu et al., 2019*). Although EC incidence and mortality rates have gradually declined, postoperative recurrence and metastasis remain the leading causes of death. Over the next decade, EC is expected to remain one of the most burdensome malignant tumors in China (*Zhou et al., 2022*).

Due to the inconspicuous early EC symptoms, most patients develop lymph node metastasis (LNM) by the time of diagnosis, resulting in poor treatment efficacy and prognosis (*DiSiena et al., 2021*). The factors affecting LNM in ESCC are unclear, but high cell proliferation may be an important contributing factor (*Deng et al., 2023*; *Wang et al., 2020*). Ki-67, expressed by the *MKI67* gene, is a nuclear antigen found in proliferating cells and is currently recognized as a marker of nuclear proliferation (*Kreipe, Harbeck & Christgen, 2022*). Ki-67 is involved in cellular synthesis and metabolism (*Lashen et al., 2023*). The degree of tumor cell malignancy can be directly reflected by its proliferative capacity, and high tumor cell proliferation is a prerequisite for infiltration and metastasis (*Menon et al., 2019*). Previous studies confirmed that Ki-67 protein is correlated with tumor occurrence, development, infiltration, metastasis, recurrence, and prognosis (*Menon et al., 2019*). Therefore, Ki-67 is considered an important parameter for prognosis and treatment selection and validated as a clinical indicator for postoperative chemotherapy for breast cancer (*Zhang et al., 2021*). Ki-67 expression is regarded as one of the most important and cost-effective alternative biomarkers for evaluating tumor cell proliferation. The Ki-67 marker index possesses prognostic and predictive value for most tumors types (*Kreipe, Harbeck & Christgen, 2022*). However, several challenges remain in routine clinical use of Ki-67, particularly the difficulty of accurate quantification in clinical settings (*Volynskaya et al., 2019*). The prognostic value of Ki-67 in ESCC remains controversial. Moreover, the relationships between Ki-67 and LNM, vascular invasion (VI), and perineural invasion (PI) in ESCC remain unclear.

The present study aimed to assess the prognostic value of Ki-67 and its relationship with LNM in patients with resected ESCC in a retrospective cohort.

## MATERIALS AND METHODS

### Patient recruitment

A total of 168 patients with stage I–IVa ESCC who consecutively received treatment at the Second Affiliated Hospital of Fujian Medical University from January 2018 to June 2023 were included in this retrospective study. The cases included in this study were patients who met the following criteria: (1) ESCC confirmed by pathology and with any type of

radical resection; (2) classified as stage I to stage IVa based on postoperative pathological examination according to the 8th American Joint Committee on Cancer (*Donohoe & Phillips, 2017*); (3) age > 18 years; (4) an Eastern Cooperative Oncology Group (ECOG) Score < 3 points or Karnofsky's performance score ≥60; (4) detailed follow-up information to calculate overall survival (OS) and (5) provision of informed consent, either from the patients themselves or their families. Exclusion criteria were: stage IVb disease; serious underlying diseases, or dual or multiple cancers.

Radical surgery was performed in all patients, with or without adjuvant treatment; however, specific surgical details were not limited. Due to the retrospective nature of this study and the potential impact of preoperative treatment on Ki-67 expression status, any patient who received neoadjuvant therapy, including chemotherapy, radiation therapy, and immunotherapy, were excluded. Given that immunohistochemical (IHC) examination of Ki-67 is routinely performed in postoperative pathology of EC in our hospital, additional collection of paraffinized tissue to assess Ki-67 expression status was not required. All IHC stained sections were obtained from Department of Pathology, and relevant clinical and pathological data were retrieved from the electronic medical record system. Written informed consent was obtained from surviving patients, or from the family members of patients who had died. Survival information for each participant was obtained from case records, telephone follow-ups, or official death records. OS was defined as the period from the date of diagnosis to the date of death from any cause or the latest follow-up. The latest follow-up period for surviving cases was up to December 31, 2023. This study was approved by the ethics committee of the Second Affiliated Hospital of Fujian Medical University (Ethics review number: 2023-416).

During the pre-established recruitment period, a total of 236 patients with ESCC underwent postoperative Ki-67 protein IHC examination. Fifty-three of these patients were excluded because they had undergone preoperative treatment. After obtaining the patients' clinical information from the electronic medical record system, 15 patients were excluded due to a lack of detailed information. Ultimately, 168 patients were included in this study. A flowchart of this study is presented in Fig. 1.

## Immunohistochemical examination and grading of Ki-67

Although additional Ki-67 immunohistochemistry experiments were not needed for this study, detailed methodological information is provided. These examinations are routinely performed at our hospital. ESCC tissue sections of four μm thickness were cut from surgically excised ESCC cancer tissue, fixed in formalin, and embedded in paraffin. All samples were mounted on silane-coated slides. IHC staining was performed using an automatic staining machine. A rabbit polyclonal antibody against Ki-67 (ab15580; Abcam) was used at 1:500 dilution for IHC staining. Two researchers simultaneously evaluated Ki-67 expression status to ensure consistent results. The intensity of Ki-67 staining was classified according to the following criteria: 0 points for no staining, 1 point for week intensity, 2 points for moderate intensity, and 3 points for strong intensity. Positive IHC-stained cells were counted for each slice under five high-power field (×200). The proportion of positive cells were classified to five groups according to their positive scores

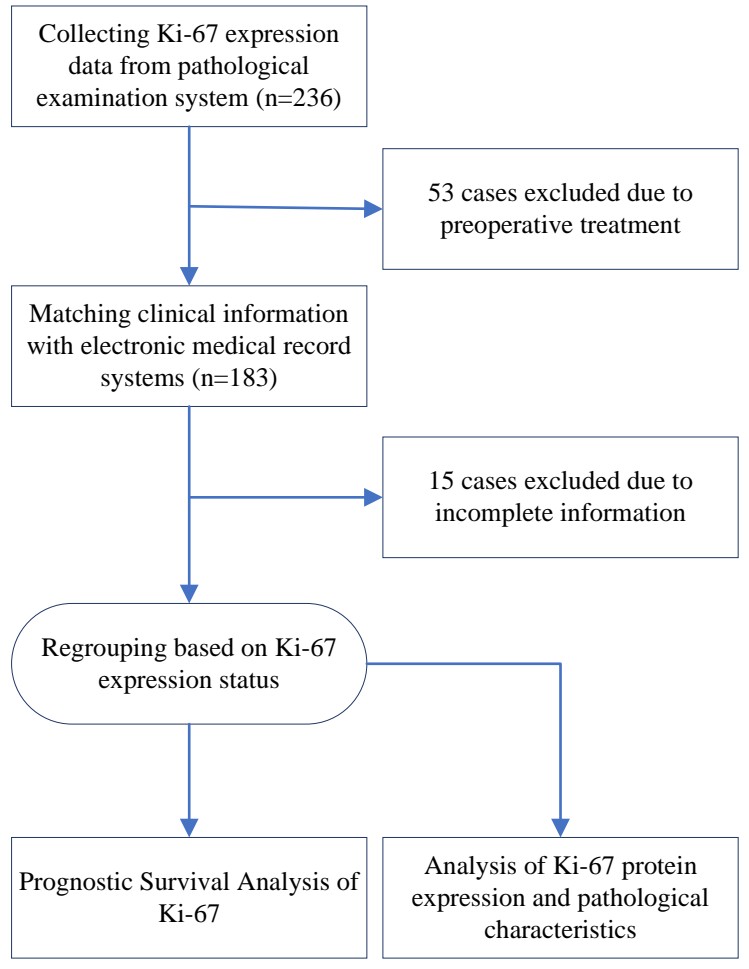

**Figure 1** Flow chart of study design.

(0 points for <5%, 1 point for 5% to 25%, 2 points for 26% to 50%, 3 points for 51% to 75%, and 4 points for 76% to 100%). The product of the intensity and positive scores was used to determine the final Ki-67 expression. Ki-67 expression scores of were divided into four groups: negative (−) for 0 points, weakly positive (+) for 1–4 points, positive (++) for 5–8 points and strongly positive (+++) for 9–12 points. The subjects were divided into a low-expression group (Ki-67$^{Low}$ group) and a high-expression group (Ki-67$^{High}$ group), with a threshold expression score of five points.

## Statistical analyses

An independent sample $t$-test was used to compare differences in continuous variables between the two groups. The chi-square test was used for categorical variables to explore the associations between Ki-67 expression and clinicopathological parameters. Propensity score matching (PSM) analysis was applied to correct for background variables and achieve statistical comparability. PSM were conducted using the "MatchIt" package in R, and the nearest neighbor algorithm is used for matching analysis. The Ki-67 expression level (high or

low) was set as the dependent variable, while other available clinicopathological parameters were set as independent variables to achieve PSM. For survival outcomes, hazard ratios (HR) and corresponding 95% confidence intervals (CIs) were used as effect sizes to show prognostic performance of Ki-67. Survival was estimated using the Kaplan–Meier method and the log-rank test. The prognostic performance of Ki-67 expression was explored using a Cox proportional hazards model. R packages "survminer" and "survival" were used to carry out survival analyses. Forest-maps based on univariate analysis and multivariate analysis from a Cox proportional-hazards model were drawn with "forestplot" R package. All statistical analyses were performed using R 4.3.1 software package *via* various R packages. Statistical significance was set at $p < 0.05$.

## RESULTS

### Clinical and pathological characteristics of the patients

A total of 168 patients with ESCC were enrolled, including 75 (44.64%) with weakly positive expression, 51 (30.36%) with positive expression and 42 (25.00%) with strong positive expression. No patients were negative for Ki-67 expression in our cohort. Finally, 75 (44.64%) patients were assigned to the Ki-67$^{Low}$ group, and 93 (55.36%) patients were assigned to Ki-67$^{High}$ group. Among these, 38 (22.62%) patients were staged at stage I, 53 (31.55%) at stage II, 65 (38.69%) at stage III and 12 (7.14%) at stage Iva; 66 (39.29%) patients were ≥60 years, and 102 (60.71%) were aged < 60 years. A total of 98 (58.33%) patients were male. Among the patients, 101 (60.12%) patients had undergone adjuvant radiotherapy or chemotherapy after surgery. The Ki-67 intensity scores were 1.31 ± 0.46 and 2.45 ± 0.50 points in the Ki-67$^{Low}$ and Ki-67$^{High}$ groups, respectively and this difference is statistically significant ($t = 15.348, P < 0.001$). The Ki-67 positivity scores were 1.95 ± 0.68 and 3.49 ± 0.50 points in the Ki-67$^{Low}$ and Ki-67$^{High}$ groups, respectively, with statistically significant difference ($t = 16.497, P < 0.001$). The Ki-67 positivity ratings were 2.56 ± 1.18 and 8.55 ± 2.07 points in the Ki-67$^{Low}$ and Ki-67$^{High}$ groups, respectively, and this difference is statistically significant ($t = 23.555, P < 0.001$). The IHC staining results of Ki-67 in the two cases of ESCC with different levels of Ki-67 expression are shown in Figs. 2A and 3A. Detailed comparative data between the two groups were shown in Table 1. Owing to significant differences in some clinical features between the Ki-67$^{Low}$ and Ki-67$^{High}$ groups, propensity matching analysis was conducted. The clinical information after matching is shown in Table 2. After PSM, the baseline characteristics between the two groups were found to be comparable ($P > 0.05$). Detailed patient data included in this study are presented in Tables S1 and S2.

### Association between Ki-67 expression and Tumor Node Metastasis (TNM) stage, lymph node metastasis, vascular invasion, and perineural invasion

The hematoxylin-eosin staining results of LNM, VI, and PI in the two cases of ESCC with different levels of Ki-67 expression are depicted in Figs. 2B, 3C, 2D, 3B, 3C and 3D. Ki-67 expression increased progressively with advancing TNM tumor stage, as shown in Table 3 and Fig. 4A. The respective proportions of patients in stages T1, T2, and T3 in the

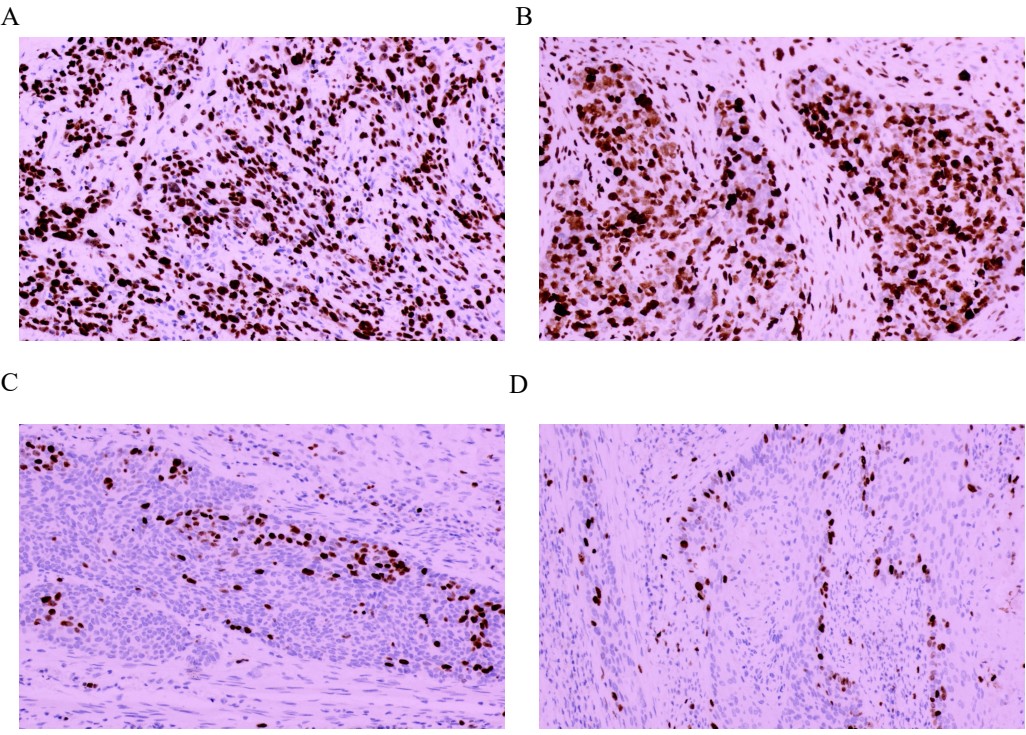

A
B
C
D

**Figure 2 Immunohistochemical staining results of Ki-67 in four cases of esophageal squamous cell cancer (×100 fields).** (A) Strongly positive expression (+++); (B) strongly positive expression (+++); (C) weakly positive expression (+); (D) weakly positive expression (+).

Ki-67$^{Low}$ group were 44.00% (33/75), 42.67% (32/75), and 13.33% (10/75), and the respective proportions of patients in stages T1, T2, and T3 in the Ki-67$^{High}$ group were 25.81% (24/93), 52.69% (49/93), and 21.51% (20/93), with a statistically significant difference ($\chi^2 = 6.468$, $P = 0.039$), as shown in Table 3 and Fig. 4B. In the entire cohort, 73.81% (124/168) of the patients exhibited lymph node metastasis. The respective proportions of patients with lymph node metastasis were 64.00% (48/75) in the Ki-67$^{Low}$ group, and 81.72% (76/93) in the Ki-67$^{High}$ group, with a statistically significant difference ($\chi^2 = 11.219$, $P = 0.011$), as shown in Table 3 and Fig. 4C. VI was observed in 35.71% (60/168) of total patients. The respective proportions of patients with VI in the Ki-67$^{Low}$ and Ki-67$^{High}$ groups were 25.33% (19/75) and 44.09% (41/93), with a statistically significant difference ($\chi^2 = 6.359$, $P = 0.012$), as shown in Table 3 and Fig. 4D. A total of 36.31% (61/168) patients had PI. The respective proportions of patients with PI in the Ki-67$^{Low}$ and Ki-67$^{High}$ groups were 24.00% (18/75) and 46.24% (43/93), with a statistically significant difference ($\chi^2 = 8.877$, $P = 0.003$), as shown in Table 3 and Fig. 4E.

## Analysis of the effect of Ki-67 expression status on ESCC prognosis

Median follow-up time was 33.5 months (range 3.0–60.0 months) in our cohort. At the end of follow-up, 63.10% (106/168) of patients had died. Kaplan–Meier survival analysis revealed OS proportions at 3 and 5 years of 74.13% (64.18%–85.62%) and

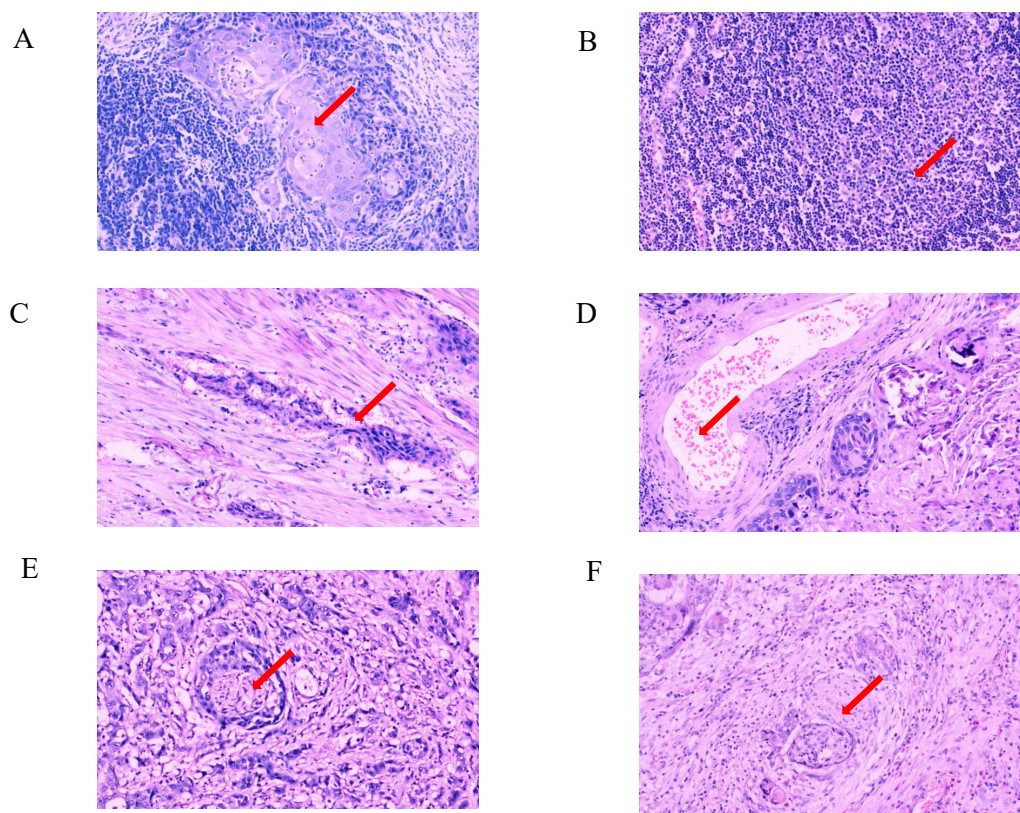

**Figure 3 Pathological characteristics in different expression status of Ki-67 (×100 fields).** (A) Hematoxylin-eosin staining of positive lymph node metastasis in one case with strongly positive expression of Ki-67; (B) hematoxylin-eosin staining of negative lymph node metastasis in one case with weakly positive expression of Ki-67; (C) hematoxylin-eosin staining of positive vascular invasion in one case with strongly positive expression of Ki-67; (D) hematoxylin-eosin staining of negative vascular invasion in one case with weakly positive expression of Ki-67; (E) hematoxylin-eosin staining of positive perineural invasion in one case with strongly positive expression of Ki-67; (F) hematoxylin-eosin staining of negative perineural invasion in one case with weakly positive expression of Ki-67.

31.55% (23.22%–42.88%) for the Ki-67$^{Low}$ group, and 51.07% (39.40%–66.20%) and 11.21% (5.81%–21.63%) for the Ki-67$^{High}$ group ($\chi^2 = 34.833$, $P < 0.001$). In the PSM cohort, Kaplan–Meier survival analysis of OS proportions at 3 and 5 years were 68.57% (56.49%–83.23%) and 33.82% (22.94%–49.86%) for the Ki-67$^{Low}$ group, and 45.22% (31.72%–64.46%) and 18.40% (9.71%–34.84%) for the Ki-67$^{High}$ group ($\chi^2 = 11.372$, $P < 0.001$). Kaplan–Meier curves for OS are depicted in Figs. 5A and 5B. The univariate Cox proportional hazards model revealed that high Ki-67 expression was significantly associated with poor OS compared to that of the low expression (HR = 3.42, 95% CI [2.22–5.27], $P < 0.001$), as shown in Table 4 and Fig. 6A.

The univariate Cox proportional hazards model also revealed that TNM stage, VI, PI and postoperative treatment exhibited significant effect on OS ($P < 0.05$). A multivariable COX model was analyzed using stepwise regression based on the first model with significant univariates in Table 4. The result of multivariable model is shown in Table 5 and Fig. 6B.

**Table 1  The baseline characteristics of patients with esophageal squamous cell cancer.**

| Parameters | Levels | Total | Ki-67 low group (n = 75) | Ki-67 high group (n = 93) | Statistics | P |
|---|---|---|---|---|---|---|
| Age | | 57.15 ± 7.65 | 57.36 ± 7.80 | 56.99 ± 7.56 | 0.312 | 0.756 |
| Age group | <60 years old | 102 (60.71) | 45 (60.00) | 57 (61.29) | 0.029 | 0.865 |
| | >=60 years old | 66 (39.29) | 30 (40.00) | 36 (38.71) | | |
| Stage | Stage I | 38 (22.62) | 24 (32.00) | 14 (15.05) | 12.989 | 0.005 |
| | Stage II | 53 (31.55) | 27 (36.00) | 26 (27.96) | | |
| | Stage III | 65 (38.69) | 22 (29.33) | 43 (46.24) | | |
| | Stage IV | 12 (7.14) | 2 (2.67) | 10 (10.75) | | |
| Post treatment | No | 67 (39.88) | 25 (33.33) | 42 (45.16) | 2.423 | 0.120 |
| | Yes | 101 (60.12) | 50 (66.67) | 51 (54.84) | | |
| Gender | Female | 70 (41.67) | 34 (45.33) | 36 (38.71) | 0.749 | 0.387 |
| | Male | 98 (58.33) | 41 (54.67) | 57 (61.29) | | |
| Tumor location | Upper thoracic | 28 (16.67) | 9 (12.00) | 19 (20.43) | 2.134 | 0.344 |
| | Mid thoracic | 80 (47.62) | 38 (50.67) | 42 (45.16) | | |
| | Lower thoracic | 60 (35.71) | 28 (37.33) | 32 (34.41) | | |
| Differentiation | G1 | 59 (35.12) | 31 (41.33) | 28 (30.11) | 2.299 | 0.317 |
| | G2 | 74 (44.05) | 30 (40.00) | 44 (47.31) | | |
| | G3 | 35 (20.83) | 14 (18.67) | 21 (22.58) | | |
| Ki-67-intensity | | 1.94 ± 0.75 | 1.31 ± 0.46 | 2.45 ± 0.50 | 15.348 | <0.001 |
| Ki-67-positive percentage | | 2.80 ± 0.97 | 1.95 ± 0.68 | 3.49 ± 0.50 | 16.497 | <0.001 |
| Ki-67-score | | 5.88 ± 3.45 | 2.56 ± 1.18 | 8.55 ± 2.07 | 23.555 | <0.001 |

In the stepwise regression multivariable model, high Ki-67 expression was significantly associated with poor OS in patients with ESCA compared to that of the low expression (HR = 1.98, 95% CI [1.33–2.94], $P < 0.001$), suggesting that high Ki-67 expression is an independent prognostic factor in patients with ESCC.

## DISCUSSION

Our present study demonstrates that high Ki-67 protein expression is associated with poor prognosis in ESCC in a real-life cohort, indicating that higher Ki-67 expression is an independent prognostic factor in patients with ESCC. Additionally, we found that increased Ki-67 expression significantly elevates the risk of LNM, VI, and PI in ESCC.

Stratifying prognosis in cancer populations based on gene expression, and exploring targeted therapies are popular topics in cancer research (*Qian et al., 2021*). Currently, cancer staging systems such as the American Joint Committee on Cancer TNM Staging System (AJCC TNM system) are primarily based on the anatomical extent of the disease (*Sudo et al., 2021*). Although grouping systems are effective and direct for most cancers, they overlook individual differences among cancer patients and fail to effectively distinguish the clinical and pathological features of specific populations (*Qian et al., 2021*). Owing to advancements in genome, transcriptome, and big data technologies, we can now explore the molecular characteristics of tumors in greater detail and assess their clinical relevance,

**Table 2  The baseline characteristics of patients with esophageal squamous cell cancer (propensity matching analysis cohort).**

| Parameters | Levels | Total | Ki-67 low group (n = 53) | Ki-67 high group (n = 53) | Statistics | P |
|---|---|---|---|---|---|---|
| Age | | 56.28 ± 7.62 | 56.62 ± 7.50 | 55.94 ± 7.80 | 0.457 | 0.649 |
| Age group | <60 years old | 68 (64.15) | 33 (62.26) | 35 (66.04) | 0.164 | 0.685 |
| | >=60 years old | 38 (35.85) | 20 (37.74) | 18 (33.96) | | |
| Stage | Stage I | 24 (22.64) | 12 (22.64) | 12 (22.64) | 0.051 | 0.997 |
| | Stage II | 37 (34.91) | 19 (35.85) | 18 (33.96) | | |
| | Stage III | 41 (38.68) | 20 (37.74) | 21 (39.62) | | |
| | Stage IV | 4 (3.77) | 2 (3.77) | 2 (3.77) | | |
| Post treatment | No | 41 (38.68) | 20 (37.74) | 21 (39.62) | 0.040 | 0.842 |
| | Yes | 65 (61.32) | 33 (62.26) | 32 (60.38) | | |
| Gender | Female | 42 (39.62) | 21 (39.62) | 21 (39.62) | 0.000 | 1.000 |
| | Male | 64 (60.38) | 32 (60.38) | 32 (60.38) | | |
| Tumor location | Upper thoracic | 13 (12.26) | 7 (13.21) | 6 (11.32) | 0.196 | 0.907 |
| | Mid thoracic | 53 (50.00) | 27 (50.94) | 26 (49.06) | | |
| | Lower thoracic | 40 (37.74) | 19 (35.85) | 21 (39.62) | | |
| Differentiation | G1 | 39 (36.79) | 20 (37.74) | 19 (35.85) | 0.417 | 0.812 |
| | G2 | 47 (44.34) | 22 (41.51) | 25 (47.17) | | |
| | G3 | 20 (18.87) | 11 (20.75) | 9 (16.98) | | |
| Ki-67-intensity | | 1.90 ± 0.74 | 1.34 ± 0.48 | 2.45 ± 0.50 | 11.684 | <0.001 |
| Ki-67-positive percentage | | 2.70 ± 0.98 | 1.92 ± 0.68 | 3.47 ± 0.50 | 13.370 | <0.001 |
| Ki-67-score | | 5.55 ± 3.40 | 2.60 ± 1.21 | 8.49 ± 2.04 | 18.026 | <0.001 |

**Table 3  Correlation analysis between Ki-67 and TNM stage, lymph node metastasis, vascular invasion, and perineural invasion.**

| Parameters | Levels | Total | Ki-67 low group (n = 75) | Ki-67 high group (n = 93) | Statistics | P |
|---|---|---|---|---|---|---|
| Stage | Stage I | 38 (22.62) | 24 (32.00) | 14 (15.05) | 12.989 | 0.005 |
| | Stage II | 53 (31.55) | 27 (36.00) | 26 (27.96) | | |
| | Stage III | 65 (38.69) | 22 (29.33) | 43 (46.24) | | |
| | Stage IV | 12 (7.14) | 2 (2.67) | 10 (10.75) | | |
| T stage | T1 | 57 (33.93) | 33 (44.00) | 24 (25.81) | 6.468 | 0.039 |
| | T2 | 81 (48.21) | 32 (42.67) | 49 (52.69) | | |
| | T3 | 30 (17.86) | 10 (13.33) | 20 (21.51) | | |
| N stage | N0 | 44 (26.19) | 27 (36.00) | 17 (18.28) | 11.219 | 0.011 |
| | N1 | 57 (33.93) | 27 (36.00) | 30 (32.26) | | |
| | N2 | 55 (32.74) | 19 (25.33) | 36 (38.71) | | |
| | N3 | 12 (7.14) | 2 (2.67) | 10 (10.75) | | |
| Vascular invasion | Negative | 108 (64.29) | 56 (74.67) | 52 (55.91) | 6.359 | 0.012 |
| | Positive | 60 (35.71) | 19 (25.33) | 41 (44.09) | | |
| Perineural invasion | Negative | 107 (63.69) | 57 (76.00) | 50 (53.76) | 8.877 | 0.003 |
| | Positive | 61 (36.31) | 18 (24.00) | 43 (46.24) | | |

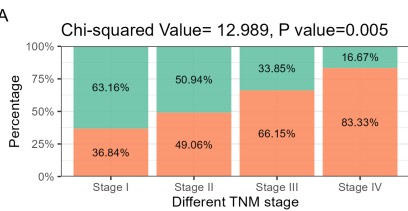

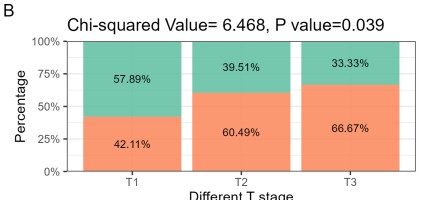

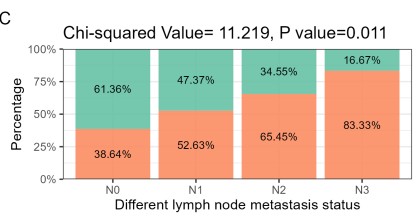

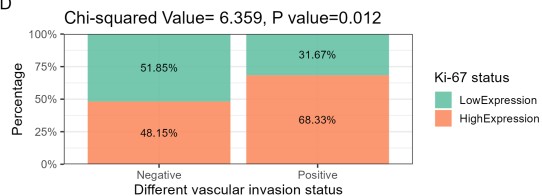

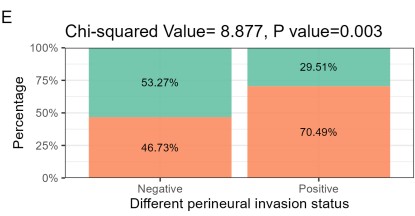

**Figure 4 Correlation analysis between Ki-67 and TNM stage, lymph node metastasis, vascular invasion, and perineural invasion.** (A) TNM stage: as tumor staging increases, the proportion of Ki-67 positive and strongly positive patients increases. (B) T stage: with the increase of T staging, the proportion of Ki-67 positive and strongly positive patients showed an increasing trend. (C) Lymph node metastasis: as the degree of lymph node metastasis increases, the proportion of Ki-67 positive and strongly positive patients increases. (D) Vascular invasion: the proportion of Ki-67 positive and strongly positive patients increases in the cohort with positive vascular invasion. (E) Perineural invasion: the proportion of Ki-67 positive and strongly positive patients increases in the cohort with positive perineural invasion.

enabling the development of more precise and efficient prognostic and predictive systems by screening different tumor subgroups for biological determinants (*Qian et al., 2021*). However, to date, only a limited number of genetic features have been incorporated for clinical staging, likely due to the challenges in determining the expression status of most genes. Elevated Ki-67 expression correlates with poor prognosis and advanced clinicopathological features in most cancers and can serve as a biomarker for disease management (*Jing et al., 2019*; *Li et al., 2021c*; *Xie et al., 2017*). To our knowledge, the detection of Ki-67 protein expression levels and status is a routine pathological examination and qualitative analysis of surgical specimens in most pathology departments, making Ki-67 protein expression a potential marker for prognosis and tumor staging (*Dokcu et al., 2023*; *Li et al., 2021b*; *Xue et al., 2020*). The present studies strongly suggests that positive Ki-67 expression is associated with tumor stage, metastasis (including both distant metastasis and lymph node metastasis), and survival (*Moltajaei et al., 2022*).

ESCC is a prevalent and often fatal malignancy worldwide, and China is experiencing a disproportionately high burden (*Yang et al., 2020a*). In China, the incidence rate of EC is

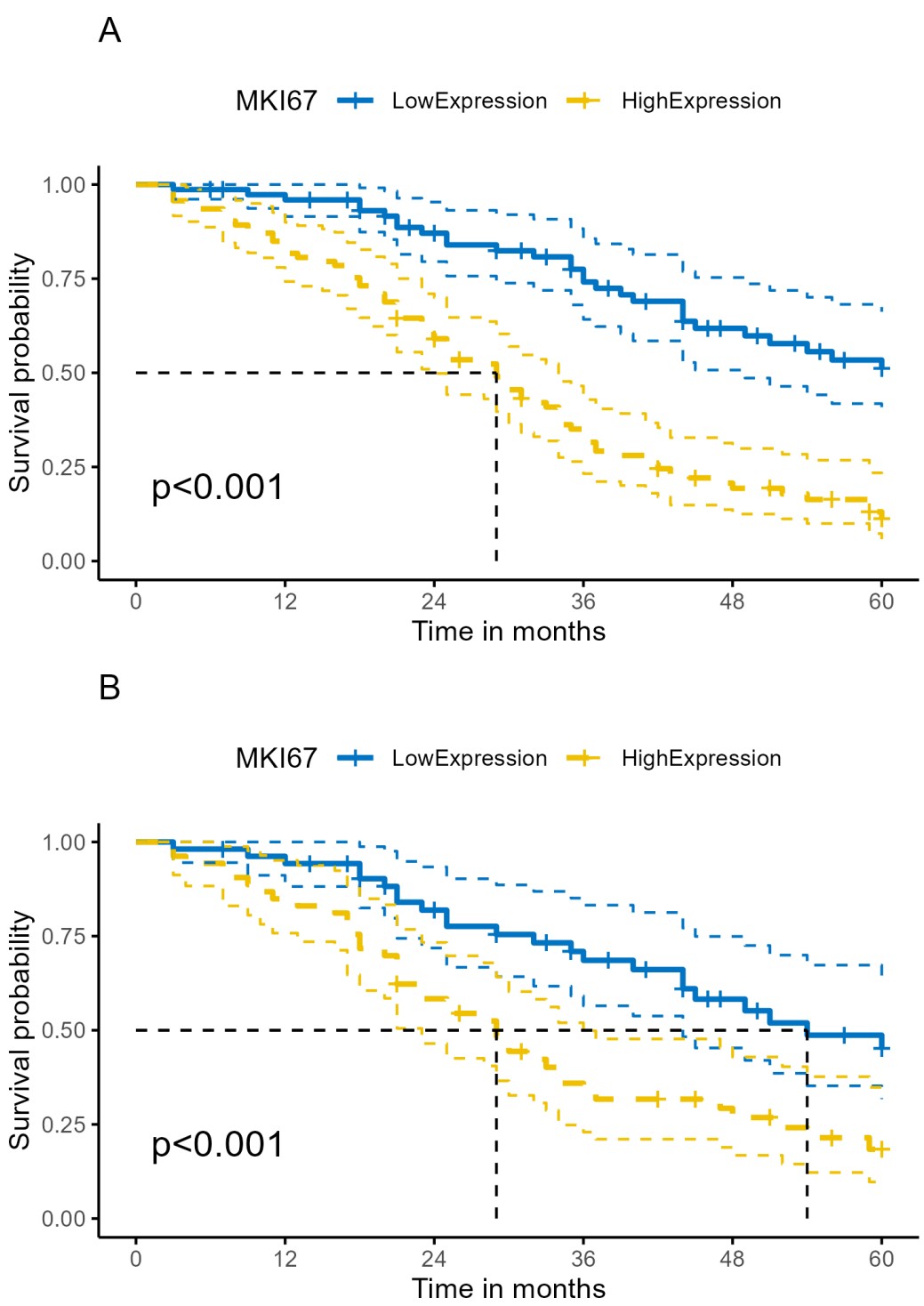

**Figure 5** **Kaplan–Meier survival analysis of Ki-67 expression on survival in human ESCC.** (A) Total cohort. (B) PSM cohort (ESCC, esophageal squamous cell cancer; OS, overall survival; PSM, propensity score matching analysis; lighter yellow or blue lines, 95% confidence interval for survival curves).

**Table 4  Univariate analysis results of overall survival in COX proportional risk model.**

| Characteristics | Levels | Beta | SE | HR (95% CI for HR) | Statistics (Z value) | P |
|---|---|---|---|---|---|---|
| MKI67 | Low expression | | | | | |
| | High expression | 1.23 | 0.22 | 3.42 (2.22, 5.27) | 5.571 | <0.001 |
| Stage | Stage I/Stage II | | | | | |
| | Stage III/Stage IV | 1.16 | 0.20 | 3.20 (2.14, 4.77) | 5.677 | <0.001 |
| Vascular invasion | Negative | | | | | |
| | Positive | 0.50 | 0.20 | 1.64 (1.12, 2.42) | 2.519 | 0.012 |
| Perineural invasion | Negative | | | | | |
| | Positive | 0.97 | 0.20 | 2.63 (1.79, 3.87) | 4.898 | <0.001 |
| Post treatment | No | | | | | |
| | Yes | −0.81 | 0.20 | 0.45 (0.30, 0.66) | −4.096 | <0.001 |
| Gender | Female | | | | | |
| | Male | −0.08 | 0.20 | 0.92 (0.63, 1.35) | −0.423 | 0.673 |
| Age group | Young group | | | | | |
| | Elderly group | 0.12 | 0.20 | 1.13 (0.77, 1.67) | 0.616 | 0.538 |
| Tumor location | Upper thoracic | | | | | |
| | Mid thoracic | −0.23 | 0.26 | 0.80 (0.48, 1.33) | −0.869 | 0.385 |
| | Lower thoracic | −0.20 | 0.27 | 0.82 (0.48, 1.40) | −0.718 | 0.473 |
| Differentiation | G1 | | | | | |
| | G2 | 0.46 | 0.23 | 1.58 (1.01, 2.48) | 2.015 | 0.044 |
| | G3 | 0.40 | 0.27 | 1.49 (0.88, 2.55) | 1.475 | 0.140 |

particularly high, with approximately 90% of cases classified as ESCC (*Zhou et al., 2022*). Although various risk factors have been identified, treatment options for patients with EC are insufficient owing to the high mutation burden of cancer and the lack of appropriate prognostic models. Our understanding of the genetic drivers of ESCC is currently limited (*Yang et al., 2020a*). At present, surgery, chemotherapy, radiotherapy, limited targeted therapy and immunotherapy yield dismal survival advantages. Thus, ESCC prognosis remains poor. Targeted therapy and immunotherapy for EC are still under investigation (*Yang, Li & Yang, 2023*; *Yang et al., 2020b*), and the screening and identification of target genes are key to the success of targeted therapy. Ki-67 expression level can serve as a key biomarker for the diagnosis of ESCC and esophageal squamous intraepithelial neoplasia, as confirmed in several studies (*Desai, 2023*; *Wang et al., 2011*).

Most studies to date have indicated that Ki-67, a tumor proliferation marker, may be associated with tumor staging. However, whether it can act as an independent prognostic indicator for patients with ESCC remains controversial (*Deng et al., 2023*). *Wang et al. (2018)* suggested that elevated Ki-67 expression is not an independent risk factor for ESCC and does not exhibit significant correlation with poor prognosis, a conclusion further supported by *Zhang et al. (2020)*. Additionally, the study by *Wang et al. (2022)*, analyzed a cohort of 226 cases and revealed that tumors with low Ki-67 indices (Ki-67 index <30%) were significantly associated with a better prognosis than that of tumors with high Ki-67 indices (Ki-67 index ≥ 30%). The same conclusion was further confirmed by *Sasagawa et al. (2012)*. However, *Deng et al. (2023)* analyzed 807 patients with ESCC and reported the
**A**

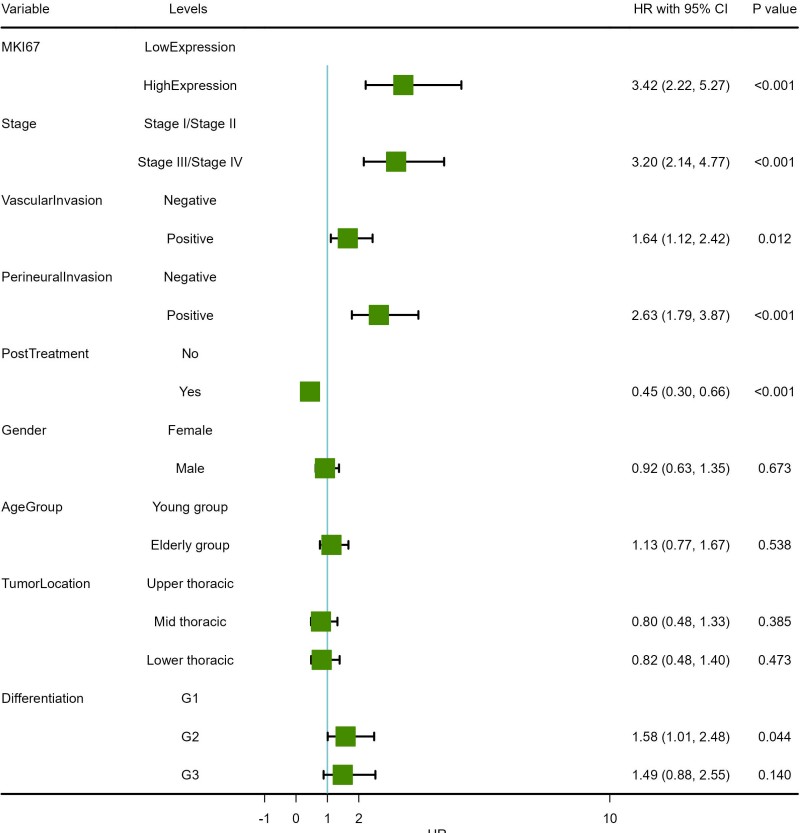

**B**

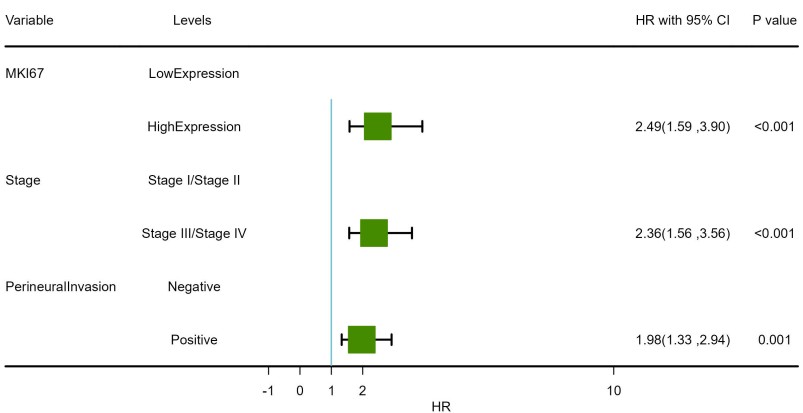

**Figure 6  Forest map of overall survival in COX proportional-hazards model in human ESCC.** (A) Results of univariate prognostic analysis. (B) Results of multivariable prognostic analysis with all significant variables. (C) Results of multivariable prognostic analysis *via* stepwise regression.

opposite conclusion. They had found that with a cut-off value of 60% for Ki-67, higher Ki-67 expression was associated with increased LNM and poor or low differentiation in patients with ESCC; however, high Ki-67 expression was not correlated with OS and disease-free survival (*Deng et al., 2023*). In our cohort, using a score of five for positive Ki-67 expression as a grouping factor, we found that Ki-67 was an independent prognostic biomarker for ESCC. There is no unified consensus on the optimal cutoff value to divide patients into groups with statistically significant differences in outcomes for Ki-67 expression, and *Deng et al. (2023)* suggested that this lack of consensus lead to varying conclusions across studies. Our univariate and multivariate results supported the prognostic value of Ki-67. The similarities and differences between several studies on Ki-67 expression are listed in Table S3. As a marker of cell proliferation, it is worth noting that Ki-67 cannot be used as a sole criterion for assessing the risk of tumor recurrence and metastasis. Other factors should be considered to develop personalized treatment plans. It is worth noting that the method reported in previous studies used the Ki-67 index, which is a continuous variable (*Sasagawa et al., 2012*; *Wang et al., 2018*). In our study, we adjusted the scoring method based on protein immunohistochemistry staining, which can effectively avoid the difficulties caused by continuous scoring. We used the intensity of Ki-67 staining and the proportion of IHC staining positive cells to comprehensively characterize the expression status of Ki-67 protein, which has also been adopted by some other scholars (*Zhang et al., 2021*). Our findings strengthen the idea that high Ki-67 protein expression is associated with poor prognosis in ESCC. However, several limitations require consideration. First, this observational study included heterogeneity in postoperative chemotherapy, radiotherapy, and immunotherapy, that may affect OS despite efforts to standardize treatment following the international guidelines for most patients within the limited sample size. Second, varying treatment strategies for tumor recurrence may exacerbated to treatment heterogeneity and affected outcomes stability. This may be an unavoidable limitation of retrospective studies. Third, this study focused solely on the clinical effect of Ki-67 on the survival prognosis of ESCC. The genetic mechanisms through which Ki-67 affects prognosis remain unclear and needs to be further investigated. Fourth, *Deng et al. (2023)*, in their study involving 807 EC patients, highlighted that Ki-67 cannot be used alone as an independent prognostic marker for EC. Although our study included a large sample size, expanding the cohort may enhance the reliability of our findings. Fifth, given the potential limitations in power and sample size, univariate and multivariable Cox proportional hazards model have not been conducted in PSM cohort. Finally, although Ki-67 is primarily associated with cell proliferation, its standalone value may not be sufficient to permit its use as a sole indicator of cancer recurrence or metastasis risk (*Green et al., 2016*). *Green et al. (2016)* suggested that Ki-67, along with DLX2 can predict increased risk of metastasis in prostate cancer. Thus, considering Ki-67 in conjunction with other metastasis-related genes may improve predictive accuracy of metastasis prediction in esophageal cancer. Further studies will focus on bioinformatics analyses to identify Ki-67-associated metastasis-related genes for integrated predictive models of esophageal cancer metastasis.

**Table 5  Multivariable analysis results of overall survival in COX proportional risk model with step-wise regression.**

| Characteristics | Levels | Beta | SE | HR (95% CI for HR) | Statistics (*Z* value) | *P* |
|---|---|---|---|---|---|---|
| MKI67 | Low expression | | | | | |
| | High expression | 0.91 | 0.23 | 2.49 (1.59, 3.90) | 3.996 | <0.001 |
| Stage | Stage I/Stage II | | | | | |
| | Stage III/Stage IV | 0.86 | 0.21 | 2.36 (1.56, 3.56) | 4.081 | <0.001 |
| Perineural invasion | Negative | | | | | |
| | Positive | 0.68 | 0.20 | 1.98 (1.33, 2.94) | 3.378 | 0.001 |

## CONCLUSION

In conclusion, high Ki-67 protein expression is associated with poor prognosis in ESCC. Increased Ki-67 expression significantly increased the risk of LNM, VI, and PI in ESCC, and may serve as an indication for adjuvant therapy.

### Funding

This study was supported by the Natural Science Foundation of Fujian Province (Grant No: 2024J01671 to Jianqing Zheng), Science and technology projects of Quanzhou city (Grant No: 2023NS010 to Jianqing Zheng); Fujian provincial health technology project (Youth Scientific Research Project, 2019-1-50 to Jianqing Zheng), the Nursery Fund Project of the Second Affiliated Hospital of Fujian Medical University (Grant No: 2021MP05 to Jianqing Zheng) and School-level Project of Quanzhou Medical College (Grant No: XJK2402A, to Bifen Huang). The funders had no role in study design, data collection and analysis, decision to publish, or preparation of the manuscript.

### Grant Disclosures

The following grant information was disclosed by the authors:
Natural Science Foundation of Fujian Province: 2024J01671.
Science and technology projects of Quanzhou city: 2023NS010.
Fujian provincial health technology project: 2019-1-50.
Nursery Fund Project of the Second Affiliated Hospital of Fujian Medical University: 2021MP05.
School-level Project of Quanzhou Medical College: XJK2402A.

### Competing Interests

The authors declare there are no competing interests.

### Author Contributions

- Jianqing Zheng conceived and designed the experiments, performed the experiments, analyzed the data, prepared figures and/or tables, authored or reviewed drafts of the article, and approved the final draft.

- Bifen Huang analyzed the data, authored or reviewed drafts of the article, and approved the final draft.
- Ying Chen analyzed the data, prepared figures and/or tables, authored or reviewed drafts of the article, and approved the final draft.
- Bingwei Zeng performed the experiments, authored or reviewed drafts of the article, and approved the final draft.
- Lihua Xiao conceived and designed the experiments, prepared figures and/or tables, authored or reviewed drafts of the article, and approved the final draft.
- Min Wu conceived and designed the experiments, authored or reviewed drafts of the article, and approved the final draft.

### Human Ethics

The following information was supplied relating to ethical approvals (*i.e.*, approving body and any reference numbers):

The study was approved by the ethics committee of the Second Affiliated Hospital of Fujian Medical University (Ethics review number: 2023-416).

### Data Availability

The raw measurements are available in the Supplementary Files.

### Supplemental Information

Supplemental information for this article can be found online at http://dx.doi.org/10.7717/peerj.19062#supplemental-information.

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
