# Peer review of "Exploratory analyses of the associations between Ki-67 expression, lymph node metastasis, and prognosis in patients with esophageal squamous cell cancer"

_PeerJ, doi:10.7717/peerj.19062_

## Round 0.1 · original submission · Major Revisions

The reviewers acknowledge the value of this paper, but also point out a number of problems, particularly with the language. Please be more precise in your wording. Also, please clearly state how this paper is novel compared to previous research.

Reviewer 1 ·

Basic reporting

As a marker of cell proliferation, the expression level of Ki67 in tumor tissue is closely correlated with the proliferation activity of tumor cells, and it is able to predict the risk of tumor recurrence and metastasis to some extent.
In this study, COX correlation analysis of KI67 in lymph node metastasis (LNM) in esophageal cancer, vascular invasion (VI) and and perineural invasion (PI) was performed, with some limitations.
It should be noted that the value of Ki67 cannot be used as the sole criterion for judging the risk of tumor recurrence and metastasis, and other factors should be considered to develop personalized treatment plans.

Experimental design

uninnovative

Validity of the findings

uninnovative

Additional comments

1.The study by Green et al. (2016) on prostate cancer suggests that KI67, along with DLX2, can predict an increased risk of metastasis formation. Although KI67 is primarily associated with cell proliferation, its value alone may not be sufficient as a sole indicator of recurrence or metastasis risk in cancer. Thus, in analyses of metastasis in esophageal cancer, it would be beneficial to consider KI67 in conjunction with other metastasis-related genes to enhance predictive accuracy.
e.g. Reference: Green WJ, Ball G, Hulman G, et al. "KI67 and DLX2 predict increased risk of metastasis formation in prostate cancer—a targeted molecular approach." Br J Cancer. 2016;115(2):236-42. doi: 10.1038/bjc.2016.169. PMID: 27336609
2.The article's language could be more precise. While both the incidence and mortality rates of esophageal cancer have shown a gradual decline, postoperative recurrence and metastasis remain the leading causes of death. Although the burden of cancer treatment in China remains substantial, esophageal cancer is not the most prominent contributor to this burden.
3.Reference 7, which involves a study with 807 patients, highlights that KI67 alone cannot be used as an independent prognostic marker for esophageal cancer. Although this study’s sample size is larger, expanding the cohort further could help increase the reliability of the conclusions.
4.In the results section, Figure 2A presents IHC staining, but "IHS" is not an appropriate term for this method. Additionally, Figures 2 and 3 could be combined for clearer comparison, making the results more concise and visually impactful.

Reviewer 2 ·

Basic reporting

While the experimental design is simple, I do find the language of the manuscript need some polishing. For example:
• Some abbreviations need to show full name/ elaboration when first appearing, e.g. TNM, T-stage
• Line 41-42: “In the long run” and “in the next 10 years” appear repetitive in one sentence
• Line 48-50: This sentence’s connection with the sentence before and after appear weak.
• Line 58: the word “scientific quantification” is a bit weird. May need to change the word of “scientific” into something else.
• Line 76: the word “double” appear weird (dual may be more suitable)

Experimental design

The experimental design is not very complex to understand, and my comments are as below:
1. Ki-67 threshold: As the authors also admit in the discussion session, the threshold may impact the result. Have the authors explored other thresholds or tried other ways to segment the patients, e.g. high, mid, low?
2. Have the authors tried to use other creative method to resolve the threshold issue, for example simply just conduct some regression between Ki-67 and other features, instead of using threshold?
3. Figure 2 and 3: Would suggest merge into one figure so easy to compare side by side. In addition the tissue structure does look very different between the 2 groups – have the authors picked the most comparable tissues? In addition, it will be best if there are more patient cases shown (not just 2 cases, 1 for each).
4. PSM: Would like to see further elaboration on how this is conduct, given this may be important point.
5. Figure 5: Why do the authors show 2 cohorts, one as total and the other as PSM? I understand PSM is needed to make the sample comparable. If that is the case I would just need to see one plot.
6. Line 187-195: Why do the authors choose to use two multivariate COX models? Some more elaboration on rationale would be better (especially for readers not familiar with the statistical methods).
7. There are already many works on the topic as cited by the authors themselves – Wang et al, Deng et al, Sasagawa et al, Zhang et al. Would be great to see a table comparing the methodology and outcome of these different studies (could be in Appendix). The authors also need to better elaborate what is new in the current study.

Validity of the findings

Overall I think the novelty of the work is a bit impacted by the several previous work on the same topic. Therefore will be great if the authors could show how this work differentiates from the previous ones (e.g. better statistical analysis?)

Reviewer 3 ·

Basic reporting

1. There is a recurring formatting issue throughout the manuscript where spacing is inconsistently applied. For example, expressions such as "HR=3.42, 95%CI= (2.22, 5.27), P<0.001)" lack consistent spacing. Proper formatting would involve consistent use of spaces, such as "(HR = 3.42, 95% CI: 2.22–5.27, P < 0.001)". Consistent spacing is critical for improving readability, ensuring clarity, and maintaining a professional presentation. I recommend carefully reviewing and correcting these inconsistencies throughout the manuscript.
2. There is a formatting issue on line 129 where a space is missing between "R 4.3.1" and "software."
3. In the Discussion section (lines 234-235), the authors state, "Wang et al. had suggested high expression of Ki-67 is an independent risk factor for ESCC and is highly correlated with poor survival." However, upon reviewing the cited paper, it is noted that the original study reported that "Ki-67 was not associated with prognosis in both OS and PFS" (https://doi.org/10.2147/OTT.S160066). The authors need to address and clarify this discrepancy.
4. Some squares and numbers (1, 2, 3) are displayed in Table 2 and Table 3, which should be removed.

Experimental design

The authors employed propensity score matching in Table 2 to enhance comparability between the two groups. However, it is unclear why the survival analysis was not performed using the matched dataset, which would offer a more balanced comparison and aligns with the purpose of conducting propensity score matching as stated in the paper.

Validity of the findings

1. The authors report that Ki-67 is significantly associated with worse overall survival in ESCA patients and repeatedly state, "high expression of Ki-67 was a significant risk factor for ESCA patients in terms of OS" (e.g., lines 27-28, 184-186, 194-195, 198-199). However, this phrasing might misleadingly suggest that Ki-67 was split into high and low groups, and separate models were run for each group, with only the high-expression model showing significance. To eliminate this ambiguity, I recommend rephrasing as: "High Ki-67 expression was significantly associated with poor overall survival (OS) in ESCA patients, comparing to low expression group."This wording accurately reflects the comparison between high and low groups without implying separate analyses.
2. In the univariate and multivariable analyses (Figure 6, Table 5, and Table 6), Stage IV shows a relatively wide confidence interval, likely due to the limited sample size in this group, with only 2 samples in the Ki-67 low group and 12 samples in total for Stage IV. To improve the balance for comparisons, the authors might consider combining Stages I and II versus Stages III and IV.

Additional comments

1. In Figure 1B, the authors include a workflow for TCGA ESCA; however, TCGA is not mentioned or discussed in the text. The authors should either provide a description of TCGA in the main text or remove it from the figure for consistency.
2. The authors describe the relationship between Ki-67 and survival outcomes using the term “correlation.” However, as noted by Altman & Krzywinski (2015), there is a distinction between "correlation" and "association." The term "correlation" specifically refers to a quantifiable measure of the strength and direction of a linear relationship between two numerical variables. In this context, the more accurate term is "association," and it should be used to describe the relationship between Ki-67 and survival outcomes.
3. In the "Statistical Analysis" section, the authors state that both the Chi-square test and Fisher's Exact test were used for comparing categorical variables. However, only the Chi-square test is mentioned in the abstract. For accuracy and consistency, both methods should be mentioned in the abstract if they were employed in the analysis.
2. Throughout the paper, the term "multivariate analysis" is used inaccurately. In this context, where multiple predictors are used to predict a single outcome (Overall Survival), the correct term is "multivariable analysis." Using the incorrect terminology could lead to confusion, as "multivariate analysis" refers to analyses with multiple outcome variables. I recommend revising the terminology across the paper to consistently use "multivariable analysis" to accurately reflect the statistical method applied.

---

## Round 0.2 · Minor Revisions

The reviewers have acknowledged the improvement of your manuscript. However, they feel that your manuscript still has some problems. I agree with them. Please revise your manuscript according to their comments.

Reviewer 2 ·

Basic reporting

The authors have improvement over the last iteration.

Experimental design

Towards the authors response:

Response #1: "The method reported in previous studies used the Ki-67 index, which is a continuous variable. In our study, we adjusted the scoring method based on protein immunohistochemistry staining, which can effectively avoid the difficulties caused by continuous scoring." As the authors also admit that the scoring method is different from other previous studies, would like to see some elaboration in the text of the paper, on why current scoring method is more superior to the method in previous studies. Same comment for response #2.

Validity of the findings

I'm fine with the response - has this been reflected in the discussion section? If so shall be ok.

Additional comments

As mentioned above, after the authors revise the above comments, it shall be ok to be accepted.

Reviewer 3 ·

Basic reporting

No comment

Experimental design

1. The authors have addressed most of the concerns. However, they present results from two multivariable models: an initial model without variable selection and another model after stepwise selection. Including both models seems unnecessary and could be misleading. I recommend retaining only the multivariable model after stepwise selection, as it is more relevant and concise. Including both models in the main text may unnecessarily complicate the presentation of your findings and could potentially confuse readers. The primary purpose of a multivariable analysis in this context is to provide a clear and concise assessment of the most important predictors, which is precisely what the stepwise regression model achieves. By focusing on the final model, you ensure that the results are more interpretable and directly applicable to clinical practice.
2. The authors provide descriptive statistics and Kaplan-Meier plots both before and after propensity score matching (PSM), but they do not show the results of a multivariable model after PSM. While this omission is acceptable given the potential limitations in power and sample size, it would be helpful to include a few sentences in the main text explaining this choice. Without such clarification, the use of the full cohort for some analyses and the PSM cohort for others could appear inconsistent.
3. The authors outline the methods for propensity score matching but should provide additional details for transparency and reproducibility. Specifically, they should clarify how the matching was conducted (e.g., nearest neighbor, caliper width, ratio of cases to controls, etc.).

Validity of the findings

no comment

Additional comments

no comment

---

## Round 0.3 · accepted · Accept

The reviewers confirmed that the authors have addressed all of the reviewer's comments. This paper is now ready for publication.

Reviewer 2 ·

Basic reporting

I'm fine with current submission and have no more comments.

Experimental design

I'm fine with current submission and have no more comments.

Validity of the findings

I'm fine with current submission and have no more comments.

Reviewer 3 ·

Basic reporting

After the authors have revised the manuscript, I have no additional comments.

Experimental design

After the authors have revised the manuscript, I have no additional comments.

Validity of the findings

After the authors have revised the manuscript, I have no additional comments.

Additional comments

After the authors have revised the manuscript, I have no additional comments.